# A Weld Pool Morphology Acquisition and Visualization System Based on an In Situ Calibrated Analytical Solution and Virtual Reality

**DOI:** 10.3390/s25092711

**Published:** 2025-04-25

**Authors:** Yecun Niu, Shaojie Wu, Fangjie Cheng, Zhijiang Wang

**Affiliations:** 1School of Materials Science and Engineering, Tianjin University, Tianjin 300072, China; 3018208192@tju.edu.cn (Y.N.); chfj@tju.edu.cn (F.C.); wangzj@tju.edu.cn (Z.W.); 2Tianjin Key Laboratory of Advanced Joining Technology, Tianjin 300072, China

**Keywords:** digital twin, analytical model, image processing, gas tungsten arc welding, virtual reality

## Abstract

A weld pool morphology acquisition and visualization system was designed in the current study, which can present real-time three-dimensional (3D) weld pool morphologies to welders. The underneath of the weld pool is calculated by utilizing an in situ calibrated analytical solution based on real-time collected welding voltage, current, and the surface boundary of the weld pool. In the meantime, the heat source distribution coefficients of the analytical solution were also calibrated through a scaling calibration method. Thus, the system updates a 3D weld pool instantaneously in weld diameter, and the error is 0.8% at the minimum, and the average value is 8.54%. Furthermore, a virtual environment was constructed by using virtual reality (VR) devices, and the visualization of the 3D weld pool model was realized by employing the hot-update technology. The experimental results demonstrate that this system is basically feasible except the update rates still need to be optimized. The current study facilitates the easier observation of weld pool morphology and is highly significant for enhancing the teleoperation skills of welders, especially in achieving precise teleoperation welding.

## 1. Introduction

Teleoperation welding has irreplaceable advantages in some special urgent welding situations, such as the repairing of nuclear pipeline leaks [1] and underwater pipeline leaks [2], and when welding in space [3], situations in which formulating an automatic welding process is time-consuming and sending a welder is impossible. This technique allows welders to perform tasks by controlling welding equipment and robots positioned remotely, with distances ranging from micrometers (for micro-manipulations) to kilometers (in space applications) [4,5]. Welders rely on teleoperated information from the welding site to assess and control the welding process. Therefore, obtaining accurate and efficient site information and presenting it appropriately to welders will significantly aid in enhancing their teleoperation skills and improving control over the welding process.

It is generally agreed in the welding community that the images/videos from the welding site contain adequate information to predict weld penetration, especially regarding the three-dimensional (3D) surface of the weld pool [6]. However, in a comparison with on-site welding, where welders can integrate various signals such as light from the weld pool [7,8,9], sound [10,11], current [12], voltage [13], and so on, Li used a binocular vision camera to construct hole identification and two-way slicing methods based on 2D images, which can detect the weld morphology after welding [14]. Gu proposed the development of binocular vision algorithms to achieve real-time monitoring of the morphology of the upper surface of the welding pool [15]. Current teleoperation welding technology can only offer the welders merely two-dimensional (2D) images and average current/voltage or the upper surface of the welding pool morphology. However, there is relatively little research on constructing the three-dimensional morphology of the underneath surface of the welding pool. Scholars around the world have employed various methods such as virtual reality (VR), augmented reality (AR), and mixed reality (MR) [16,17], as well as Digital Twins [18,19], to increase the variety and quantity of the on-site welding information provided to the welders. Qiyue Wang utilized deep learning technology to successfully monitor and control complex manufacturing processes [20]. S. J. Chen provided the optimal welding speed to the welder to instruct the welder to carry out the welding by recording the moving speed of the welding torch, which was integrated with a vibration sensor on the welding helmet [21]. In addition, Li proposed a novel method using the LSTM neural network to achieve the measurement of the 3D weld pool upper surface [22]. Xu explored a combination of two types of network strategies (computer stereo vision and classification decision) and adopted the supervised learning method to optimize the model for identifying the penetration state of the V-groove [23]. Companies such as Fronius [24] and Miller Electric [25] have created a comprehensive training space for rookie welders by integrating VR, AR, and various sensors. However, most of the current technologies cannot be applied in teleoperation welding because the on-site welding information transmitted back to the remote-control region is not enough to assist the skilled welders in handling tasks accurately, especially in terms of reconstructing the underneath surface of the three-dimensional weld pool.

In this paper, a weld pool morphology acquisition and visualization system was designed, which can provide a 3D weld pool to the welders through VR glasses. The organizational structure of this paper is as follows: In Section 2, an overview of the relevant equipment and technologies used in this research is introduced. In Section 3, the methods for acquiring and visualizing the morphology of the weld pool are presented. This is further elaborated in two parts: the 3D reconstruction process and the 3D model visualization. In Section 4, relevant verification experiments are carried out to determine the feasibility of the system. Finally, conclusions are drawn in Section 5.

## 2. System Design

The proposed weld pool morphology acquisition and visualization system contains two working regions: the on-site region and the remote-control region, as shown in Figure 1. A brief introduction to the core algorithms of 3D reconstruction and data transformation is also displayed in Figure 1. The on-site region is the actual welding scenario, including equipment such as a welding robot, a welding machine, a high dynamic range (HDR) camera, and data acquisition equipment for the welding current and voltage, as shown in Figure 1a. The remote-control region contains hardware devices such as a VR headset and an industrial control computer, as shown in Figure 1c. The virtual environment is developed by using the Unity software, where the 3D weld pool and welding current/voltage can all be presented in front of the welder, while the welding current can also be adjusted through the VR controller.

The algorithms for 3D reconstruction and data transformation are shown in Figure 1b. The weld pool images were captured by the HDR camera with a frequency of 30 Hz, and the welding voltage and current were collected by data acquisition equipment with a frequency of 10,000 Hz. The interval time between each processing program of the 3D reconstruction and visualization is set as the moment ($t_{0},\;t_{1},\cdots ,t_{n}$). A 3D model of the weld pool will be obtained during each processing program through the calculation of an analytical model by using the weld pool images and the average values of welding voltage and current, which is the arithmetic mean of every 10,000 sets of welding voltage and current. And then, the 3D model and the average value of welding voltage and current are packaged together to the remote-control end, where the 3D model can be observed by the welders.

The detailed single processing program of 3D weld pool reconstruction and visualization is shown in Figure 1d. As the original weld pool image and the mean current and voltage are obtained, image processing is proceeded firstly to achieve the boundary size of the weld pool, and then an analytical solution is calculated and calibrated by the welding current, voltage, and the size of the weld pool to obtain the underneath boundary of the weld pool. A reconstruction algorithm will subsequently rebuild the 3D weld pool. However, the reconstructed 3D weld pool model cannot be presented in the VR glass unless the format is corrected. Therefore, the 3D model will be transmitted to the virtual environment through the processes of model format conversation, model data transmission, and model instantiation, such that the 3D weld pool and welding voltage/current can be displayed on the user interface (UI). Instead of the simple 2D images directly transmitted into the UI interface, our method will provide welders with a 3D weld pool model that can be freely observed from multiple angles, which significantly enhances welders’ intuitive understanding and precise control of the welding process.

## 3. The 3D Reconstruction Process and Model Visualization

### 3.1. The 3D Reconstruction Process

The details of the 3D reconstruction process are shown in Figure 2. The processing program time ($t_{0},\;t_{1},\cdots ,t_{n}$) is taken to illustrate this 3D reconstruction process, in which the welding current is changed to explain the process in detail. In this section, the welding current before $t_{0}$ is uncertain and remains constant from $t_{0}$ to $t_{1}$, and then changes from $t_{1}$ to $t_{2}$, and whether the heat source parameters $\left(r_{h},c_{h}\right)$ need to be corrected or not is based on the welding current. The calculation and calibration of the weld pool analytical solution at $t_{0}$ is shown in Figure 2a. It is necessary to calibrate at $t_{0}$ because it is unknown whether the $\left(r_{h0},c_{h0}\right)$ is fitted to the current ($I_{0}$) or not. A preliminary weld pool’s diameter ($d_{com-p0}$) and penetration depth ($h_{com-p0}$) will be obtained through the analytical model calculation and matched with the diameter ($d_{mea-0}$) of the weld pool obtained by image processing; the $\left(r_{h0},c_{h0}\right)$ will then be updated as $\left(r_{h1},c_{h1}\right)$ through a calibration algorithm. Finally, the accurate weld pool’s diameter ($d_{0}$) and penetration depth ($h_{0}$) will be obtained by using the $\left(r_{h1},c_{h1}\right)$. Since the welding current ($I_{1}$) remains unchanged ($I_{1}=I_{0}$) during the period ($t_{0}–t_{1}$), it can be assumed that the previous $\left(r_{h1},c_{h1}\right)$ are still accurate, allowing for the determination of the weld pool’s diameter ($d_{1}$) and penetration depth ($h_{1}$) at $t_{1}$ without the calibration of $\left(r_{h1},c_{h1}\right)$, as shown in Figure 2b. The welding current ($I_{2}$) changes ($I_{2}\neq I_{1}$) continuously during the period ($t_{1}$–$t_{2}$), the calibration of the $\left(r_{h2},c_{h2}\right)$ is needed once again, and the weld pool diameter ($d_{2}$) and penetration depth ($h_{2}$) at moment ($t_{2}$) can be determined, as shown in Figure 2c. By continuing this process until $t_{n}$ (Figure 2d), the weld pool diameter ($d_{n}$) and penetration depth ($h_{n}$) can be obtained at any moment ($t_{0},\;t_{1},\cdots ,t_{n}$).

The calculation formulas of the analytical solution for calculating the welding temperature distribution over the time (0$,t_{k+1}$) are shown in Equations (1)–(3), which are deduced from our previous research [26]. Equation (1) is the overall mathematical analytical model, which can be decomposed into two different integrals with respect to time, (0$,\;t_{k}$) and ($t_{k},t_{k+1}$). Both $t_{k}$ and $t_{k+1}$ are any moment within the time range composed of ($t_{0},\;t_{1},\cdots ,t_{n}$). These integrals, respectively, correspond to the welding temperature field $T{\left(x,y,z\right)}_{t_{k}}$ obtained for the previous time (0$,\;t_{k}$) (Equation (2)) and the welding temperature field generated by the new heat input power $P_{t_{k}}$ during ($t_{k},t_{k+1}$) (Equation (3)).(1)$$T{\left(x,y,z\right)}_{t_{k}+1}=A\left(T{\left(x,y,z\right)}_{t_{k}}\right)+B_{x,y,z}Q_{t_{k}}$$
(2)$$\begin{array}{c} A\left(T(x,y,z{)}_{t_{k}}\right)=T(x,y,z{)}_{t_{k}}*g(x,y,z{)}_{\Delta t}=\int_{0}^{t_{k}}\;\frac{3\sqrt{3}Q\left(\tau \right)}{\rho c\pi \sqrt{\pi }}.\\ \frac{1}{\sqrt{12a\left(t_{k}+\Delta t-\tau \right)+a_{h}^{2}}\sqrt{12a\left(t_{k}+\Delta t-\tau \right)+b_{h}^{2}}\sqrt{12a\left(t_{k}+\Delta t-\tau \right)+c_{h}^{2}}}\\ \cdot \exp\left(-\frac{3x^{2}}{12a\left(t_{k}+\Delta t-\tau \right)+c_{h}^{2}}-\frac{3y^{2}}{12a\left(t_{k}+\Delta t-\tau \right)+a_{h}^{2}}-\frac{3z^{2}}{12a\left(t_{k}+\Delta t-\tau \right)+b_{h}^{2}}\right)d\tau \\ =\int_{0}^{t_{k}}\;Q\left(\tau \right)\cdot f_{x,y,z}\left(t_{k}+\Delta t-\tau \right)d\tau \end{array}$$
(3)$$\begin{array}{c} B_{x,y,z}Q_{t_{k}}=\int_{0}^{\Delta t}\;\frac{3\sqrt{3}Q_{t_{k}}}{\rho c\pi \sqrt{\pi }}.\\ \frac{1}{\sqrt{12a\left(\Delta t-\tau \right)+a_{h}^{2}}\sqrt{12a\left(\Delta t-\tau \right)+b_{h}^{2}}\sqrt{12a\left(\Delta t-\tau \right)+c_{h}^{2}}}\\ \cdot \exp\left(-\frac{3x^{2}}{12a\left(\Delta t-\tau \right)+c_{h}^{2}}-\frac{3y^{2}}{12a\left(\Delta t-\tau \right)+a_{h}^{2}}-\frac{3z^{2}}{12a\left(\Delta t-\tau \right)+b_{h}^{2}}\right)d\tau \\ =\int_{0}^{\Delta t}\;Q_{t_{k}}\cdot f_{x,y,z}\left(\Delta t-\tau \right)d\tau \end{array}$$


In the equation, (*a*) is the thermal diffusivity, a=λ∕ρc, (λ) is the thermal conductivity of the workpiece, (*c*) is the specific heat capacity at a constant volume, and (ρ) is the density of the workpiece. The $g{\left(x,y,z\right)}_{\Delta t}$ represents the temperature distribution under the distributed heat source [27].

The authors of Ref. [28] have explored the relationship among the calibrated $\left(r_{h},c_{h}\right)$, the measured diameter ($d_{\mathrm{mea}}$), and the measured elevation volume ($V_{\mathrm{mea}\;}$) of the weld pool. However, only the $d_{\mathrm{mea}}$ is corrected due to the inability to obtain the weld pool convexity volume in this study. The $d_{mea}$ is obtained through the image processing algorithm as shown in Figure 2f, which contains steps such as grayscale processing, binarization, and boundary extraction. Considering that the welding heat input used in this study differs from that in previous research, the relevant coefficients in the calibration formula are recalculated. There was a total of 108 sets of heat source parameters used in order to recalculate the coefficients in the formula, and the results are partially given in Table 1. The range of $r_{h}$ was 0.2 to 5 and $c_{h}$ was 0.2 to 1.2. Both of the incrementals were 0.2; the welding heat input (Q) was set to 500 W, 600 W, and 700 W, resulting in 25 × 6 × 3 = 450 sets.

A total of 334 sets of valid data were finally obtained after removing the invalid data. The fitted equation is shown in Equations (4a)–(4c) and (5). Equation (4a)–(4c) describes the relationship between the $d_{\mathrm{mea}}$ and the $\left(r_{h},c_{h}\right)$, and Equation (5) is used to calculate the optimal solution.(4a)$$d_{\mathrm{com}-500\;}=2.6971+0.2373\cdot r_{h}+0.0355\cdot c_{h}-0.0742\cdot r_{h}\cdot c_{h}-0.2254\cdot r_{h}^{2}-0.1114\cdot c_{h}^{2}$$(4b)$$d_{\mathrm{com}-600\;}=2.7361+1.8911\cdot r_{h}+0.4312\cdot c_{h}-0.2787\cdot r_{h}\cdot c_{h}-0.8958\cdot r_{h}^{2}-0.2733\cdot c_{h}^{2}$$(4c)$$d_{\mathrm{com}-700\;}=3.7632+0.9868\cdot r_{h}+0.1379\cdot c_{h}-0.1344\cdot r_{h}\cdot c_{h}-0.4364\cdot r_{h}^{2}-0.1173\cdot c_{h}^{2}$$

The MSE was 0.000759 mm^2^ for $d_{\mathrm{com}-500\;}$, 0.10147 mm^2^ for $d_{\mathrm{com}-600\;}$, and 0.039151 mm^2^ for $d_{\mathrm{com}-700\;}$. The optimal solution can be described as follows:
(5)$$\underset{r_{h},c_{h}}{min}\;J\left(r_{h},c_{h}\right)={\left(d_{com}\left(r_{h},c_{h}\right)-d_{\mathrm{mea}}\right)}^{2}$$


The 3D models of the weld pool in the format of (.obj) are obtained after the above work is completed. Next, it is necessary to transmit the 3D models to the VR headset worn by the welder to achieve the 3D model visualization.

### 3.2. The 3D Model Visualization

The objective of the 3D model visualization is to display the 3D weld pool model achieved in Section 3.1 within the remote-control region. The Unity software (Version 2020) is used to build a virtual environment for welders to observe, which contains the welding plate, weld pool, and the welding torch. However, the virtual environment setup, such as the welding materials and the 3D weld pool models, can only operate before the program runs (compilation mode), which makes the prompt updating of the 3D weld pool become difficult. Therefore, the hot-update technology, which supports the real-time modification of 3D models during operating mode without restarting the virtual environment, is adopted, as shown in Figure 3.

Nevertheless, the model format needs to be converted to meet Unity’s requirements before the hot update. The objective of the format conversion is to convert the original 3D model (M_O_) in the .obj format into a 3D model (M_F_) in the .fbx format, which is compatible with the Unity software, as shown in Figure 3a, and which can be achieved using the Blender (Version 3.0) and Python (Version 3.7) software.

The hot-update process contains three parts: the server side (Figure 3b), which is responsible for receiving and uploading the M_F_; the cloud side (Figure 3c), which is responsible for temporarily storing the M_F_; and the client side (Figure 3d), which is responsible for setting up the corresponding virtual environment to achieve the visualization and interactivity of the M_F_. Both the server side and the client side are independent Unity application programs, and the cloud side is a digital cloud built using Serve-U, which is a platform that facilitates resource sharing and transmission.

The MD5 technology is used to encrypt the M_F_ on the server side, which is a widely used cryptographic hash function that can ensure the consistency of information during transmission; a 128-byte array will be generated after applying the MD5 encryption algorithm, which ensures the stability of the M_F_ during the transmission process.

The AssetBundle (AB) technology is used to package and generate the corresponding AB files (file 1 (.ab), file 2 (.ab), …, file n (.ab)) and the resource comparison file (RCF). The RCF records detailed information such as the number, names, and version numbers of the M_F_. The M_F_ in the format of .ab and the RCF in the format of .txt are packaged together and uploaded to the cloud side via the TCP data stream on the server side, and the RCF will generate the download list, which contains the M_F_ needing to be updated, deleted, or newly added. Finally, the $M_{FC}$ is presented to the welders on the client side.

The virtual environment device we used in the experiment is the HTC Vive Pro, which has an update frequency of 90 Hz. Its display delay time is negligible compared with the delay times of analytical calculations and data transmissions.

The virtual environment on the client side contains two parts: the UI and the welding plate, as shown in Figure 4. The UI is located directly in front of the welding plate for the welder to observe easily, which contains the I and welding voltage $\left(U\right)$, as shown in Figure 4a. The $M_{FC}$ will directly appear within the weld seam on the welding plate during the welding process, as shown in Figure 4b. The functions of controlling the start and end of welding, as well as adjusting the I, are bound to the VR controller, as shown in Figure 4c. The two sides of the circular disc in the middle of the VR controller are used to control the increase and decrease in the welding current, respectively, and the central area of the disc controls the start/stop of the welding.

## 4. Experiment

The feasibility of the proposed 3D weld pool morphology acquisition and visualization system was validated through a fixed-point welding experiment. The materials used in the experiments were 304L stainless steel plates, with dimensions of 170 mm × 165 mm × 7 mm. The physical parameters of the materials used in the analytical calculations are shown in Table 2, which is sourced from [27]. The analytical model of the welding temperature field is calculated in real time by a computer configured with an Intel Core i9 processor, 16 GB of running memory, and an Nvidia GeForce GTX 1060 graphics card.

The experimental process is as follows: Perform continuous welding 5 times, with the welding duration of each time being 90 s. The current changes once every 30 s. The designed and measured welding currents $\left(I\right)$ during welding process Ⅰ are shown in Figure 5. The acquisition frequency of the current is 10,000 Hz, and the maximum error is approximately 2A. The average values of the measured current are consistent with the design values, which are used for the calculation of the analytical solution. The remaining several welding processes are similar.

The *I* during the welding process and other relevant data, including the welding heat input (*Q*), the welding heat source parameters $\left(r_{h},c_{h}\right)$, the welding depth from calculation ($h_{com}$), and the weld pool diameter ($d_{mea}$) used for calibrating the $\left(r_{h},c_{h}\right)$ in each period, are shown in Table 3. IDs I to Ⅴ are the positions where the five welding tests were conducted. The weld pool diameters ($d_{mea}$) used in the calibration are obtained from the weld pool images. The $\left(r_{h},c_{h}\right)$ will be calibrated every 30 s with the changes in *I*.

The appearance of the weld seam is shown in Figure 6a. IDs I to Ⅴ are the positions where the five welding tests were conducted. The metallographic images are shown in Figure 6b; the diameter (d) and depth (h) of the weld pool in each image are marked and compared with the corresponding ($d_{com}$, $h_{com}$) calculated during the 3D reconstruction process, as shown in Table 4. It can be found that the error of the $d\;\mathrm{and}\;d_{com}$ is 0.8% at the minimum and 24.3% at the maximum, and the average value is 8.54%. The error of the h and $h_{com}$ is 13.3% at the minimum and 30.0% at the maximum, and the average value is 24.1%, which is mainly because the convexity volume was not obtained as a correction term during the correction process and will be optimized in future work. The average processing time (5 s) for calculating and calibrating the 3D model does not yet meet the requirements (within 1 s) for actual teleoperation welding, which is due to the complexity of the mathematical analytical model this study used and the insufficient performance of our hardware equipment. The issue will be solved by improving the computational speed of the analytical model and using higher-performance hardware devices in subsequent studies, thereby decreasing the time required for model construction and transmission to an acceptable level.

The virtual environment designed within the VR headset, along with the weld seam that changes over time, is shown in Figure 7. The physical scenario is shown in Figure 7a when the welder is operating. The virtual environment can be observed through the welding helmet. The scene on the computer is captured to display the virtual environment since the scene in the welder’s helmet cannot be directly observed, as shown in Figure 7b. The I and U are displayed on the UI, and the $M_{FC}$ and the weld seam can be observed in the scene as well. The gradually lengthening weld seam is shown in Figure 7c. It can be found that the weld seam gradually lengthens, and the size of the 3D models also continues to increase as time changes, which is caused by the continuous increase in the I used in the experiment over time.

In addition, welders in the virtual environment can move their position and change their viewpoint to observe the weld pool from different angles, obtaining sufficient information about the weld pool’s morphology; the weld pool observed by the welder from different perspectives is shown in Figure 8. The delay in transferring the 3D model to the VR headset worn by the welders is approximately 2 s, which is mainly because of the hot update. Optimization can be achieved by improving the performance of the hardware equipment. The process of 3D model visualization does not affect the accuracy of the model because the 3D model is transmitted directly to the VR headset with only modifications in terms of format, which proves the feasibility of the hot-update technology in the process of 3D model visualization. However, the rendering fineness of the model is still at a relatively low level in terms of the current situation, which means that there is still room for improvement in the visual presentation effect of the model, and the tip of the welding torch did not exhibit the corresponding arc phenomenon during the welding process. All the above situations belong to the deficiencies in the performance aspect and can be optimized in a targeted manner in subsequent in-depth research.

## 5. Discussion

Accuracy and time delay are two factors of the weld pool morphology acquisition and visualization system. In terms of accuracy, the precision of analytical calculation is a significant factor affecting the system’s accuracy, while other processes, including the format conversion, packaging, and updating of the model, have no obvious impact on the model’s accuracy. The average error of the three-dimensional model can be optimized by improving the analytical model itself or by using more accurate data processing methods during the process of correcting the analytical model. For example, the deep learning network can be used to obtain a more accurate shape of the welding pool when processing the welding pool images. It will be of greater help for welders to judge the teleoperated welding process if the error of the three-dimensional model can be controlled within 10%.

In terms of time delay, there are two main factors in the weld pool data acquisition and visualization system that can lead to an increase in time delay. Firstly, the calculation time of the analytical model, which is approximately 5 s because the iterative time is set to 5 s (Δt in Equations (2) and (3)) considering the processing time of other processes. Secondly, the time spent on model packaging and transmission (2 s), which is mainly related to the performance of the hardware equipment. The time delay can be reduced to within 1 s by using the latest graphics card, RTX 5090, whose computing performance is 20 times that of the equipment used in this paper (RTX 1060). The delay time of the entire system is taken as the longer value of the above two types of time delays by running the above two processes in parallel, which is about 5 s. The delay time can be optimized by improving the performance of the hardware equipment or decomposing the calculation process into multi-threaded synchronous calculations and using more computing devices for parallel computing.

## 6. Conclusions

This study designed a weld pool morphology acquisition and visualization system. The major findings can be summarized as follows:(1)The system has successfully applied the existing welding analytical model in practice and optimized the algorithms of the image processing and analytical model calculations in order to achieve the requirements of real-time feedback. In real welding experiments, it has been proved that the error in the welding diameter is 0.8% at the minimum, and the average value is 8.54%. The welding experiments have proven the feasibility of the analytical model and the necessity of calibrating the heat source distribution coefficient.(2)The hot-update technology is used to transmit the three-dimensional model to the virtual environment, including the conversion of the model format before transmission, the use of the AssetBundle technology to package the resource files during the transmission process, and the instantiation of the model in the virtual environment. Experiments have proved that this process is completely feasible and has no impact on the accuracy of the model. The overall transmission time is 2 s, which is mainly due to the limitations of the hardware devices.

## Figures and Tables

**Figure 1 sensors-25-02711-f001:**
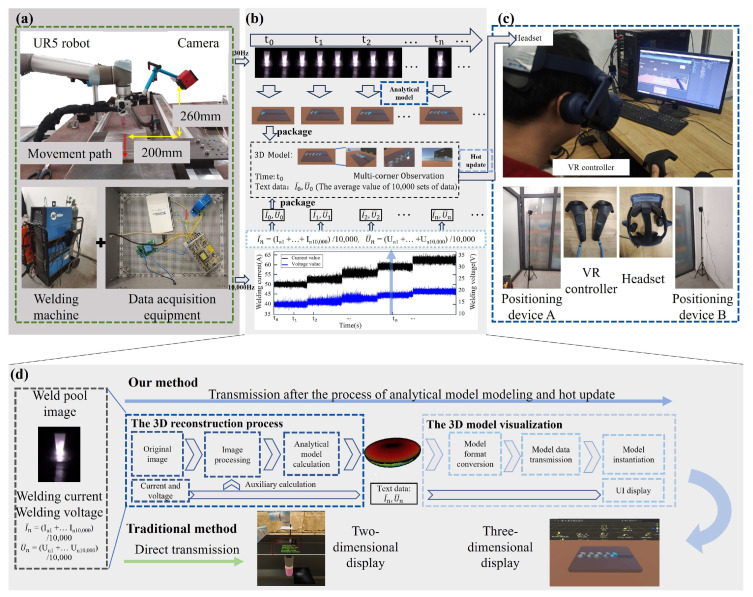
A framework diagram of the weld pool morphology acquisition and visualization system. (**a**) On-site region, which is represented by a dark gray background; (**b**) 3D reconstruction and data transformation, which is represented by a light gray background; (**c**) Remote-control region, which is represented by a light gray background; (**d**) The detail of 3D reconstruction and data transformation, which is represented by a light gray background.

**Figure 2 sensors-25-02711-f002:**
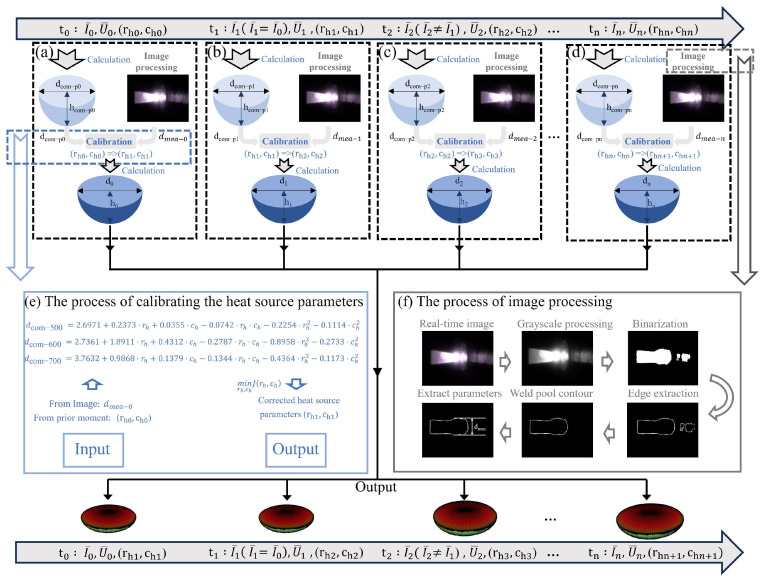
A framework diagram of the 3D reconstruction process; (**a**) The calculation process of the welding pool model at the moment of $t_{0}$; (**b**) The calculation process of the welding pool model at the moment of $t_{1}$; (**c**) The calculation process of the welding pool model at the moment of $t_{2}$; (**d**) The calculation process of the welding pool model at the moment of $t_{n}$; (**e**) The process of calibrating the heat source parameters, which is represented by the blue arrow and square frame; (**f**) The process of image processing, which is represented by the gray arrow and square frame.

**Figure 3 sensors-25-02711-f003:**
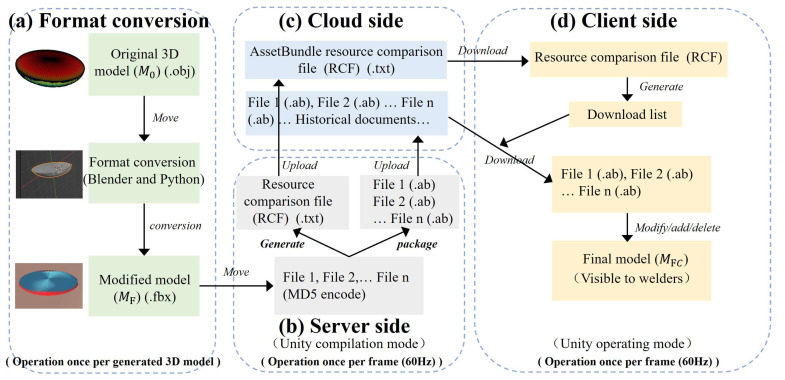
The process of the hot update; (**a**) Format conversion; (**b**) Server side; (**c**) Cloud side; (**d**) Client side.

**Figure 4 sensors-25-02711-f004:**
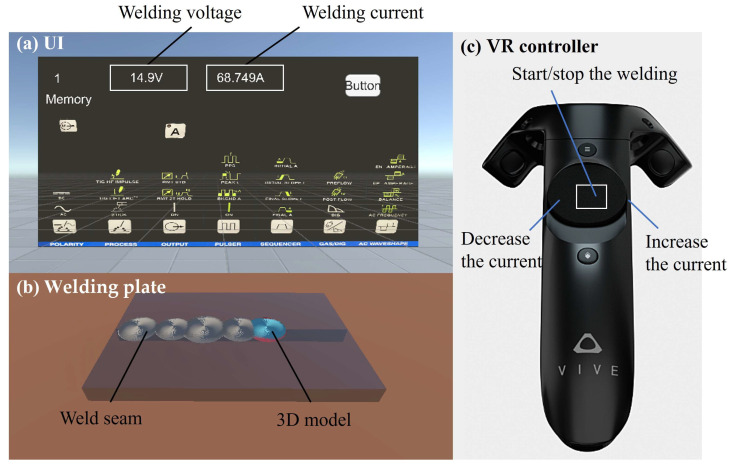
The virtual environment on the client side. (**a**) The UI for displaying the voltage and current. (**b**) The plate for displaying the weld seam and 3D model. (**c**) The VR controller for displaying the control buttons.

**Figure 5 sensors-25-02711-f005:**
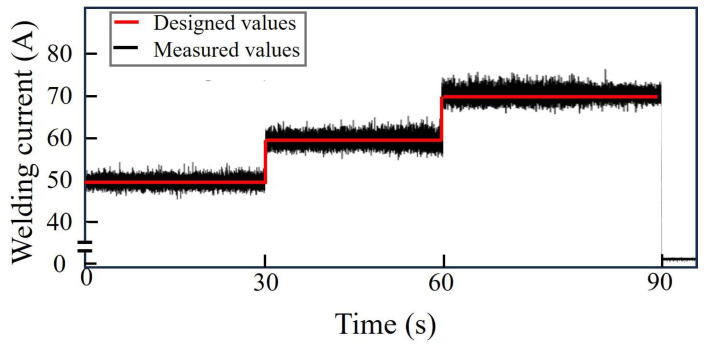
The designed values and measured values of the welding current in welding process I.

**Figure 6 sensors-25-02711-f006:**
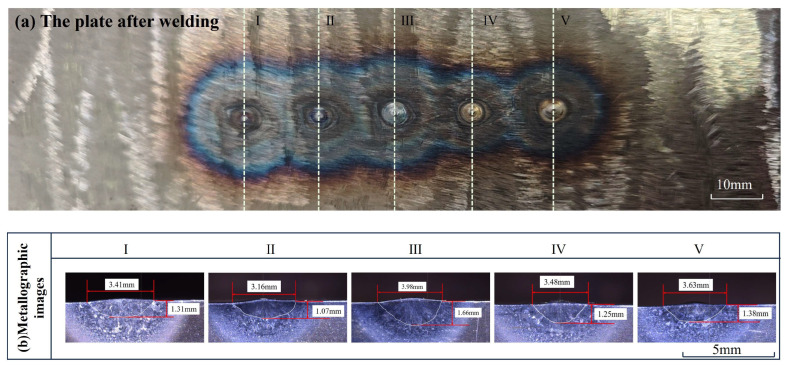
(**a**) The shape of the weld after the completion of welding; (**b**) actual metallographic images after welding.

**Figure 7 sensors-25-02711-f007:**
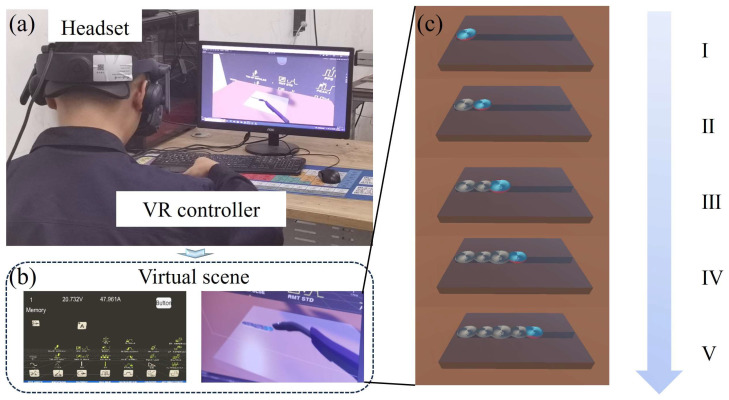
Interface diagram of the actual operation and observation by the welder during the welding process; (**a**) The physical scenario; (**b**) The scene on the computer; (**c**) The gradually lengthening weld seam.

**Figure 8 sensors-25-02711-f008:**
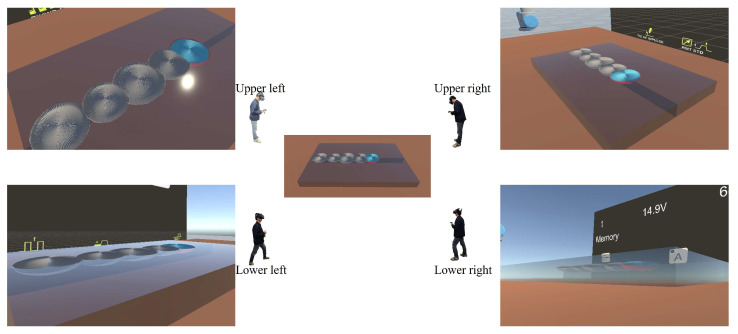
Images of the weld pool from different perspectives.

**Table 1 sensors-25-02711-t001:** Analytic model-computed ($d_{com}$).

ID	Q (W)	$r_{h}$	$c_{h}$	$d_{com}$ (mm)
1	500	0.2	0.2	2.8
2	500	0.2	0.4	2.72
…	…	…	…	…
448	700	5.0	0.8	0
449	700	5.0	1.2	0
450	700	5.0	1.2	0

**Table 2 sensors-25-02711-t002:** Physical parameters of the materials.

Input Parameters	Set Values
Initial welding temperature (°C)	0
Density ρ (kg·mm^−3^)	8.03 × 10^−6^
Specific heat capacity c (J·kg·K^−1^)	500
Thermal conductivity λ (W·mm·K^−1^)	16.2 × 10^−3^
Thermal diffusivity a (mm^2^·s^−1^)	4.035
Volumetric expansion coefficients α_v_ (K^−1^)	4.5 × 10^−5^
Melting point (°C)	1400

**Table 3 sensors-25-02711-t003:** Process parameters related to the trial results.

ID	Time (s)	Welding Current (I) (A)	Welding Heat Input (Q) (W)	Diameter from Calculation ($d_{com}$) (mm)	Heat Source Parameters ($r_{h},c_{h}$)
I	0–30	50	500	2.11	(2.020, 0.743)
	30–60	60	600	2.70	(2.008, 0.696)
	60–90	70	700	3.37	(2.445, 0.701)
II	0–30	50	500	2.14	(1.998, 0.693)
	30–60	60	600	2.73	(1.992, 0.695)
	60–90	50	500	2.95	(0.493, 0.200)
III	0–30	70	700	3.63	(2.224, 0.697)
	30–60	60	600	3.46	(1.475, 0.69)
	60–90	70	700	3.93	(1.873, 0.746)
IV	0–30	70	700	3.49	(2.350, 0.702)
	30–60	50	500	2.98	(0.493, 0.200)
	60–90	60	600	3.40	(1.535, 0.693)
V	0–30	70	700	3.71	(2.145, 0.697)
	30–60	60	600	3.46	(1.535, 0.693)
	60–90	60	600	3.41	(1.526, 0.689)

**Table 4 sensors-25-02711-t004:** The values (*d*, *h*) and the corresponding values ($d_{com}$, $h_{com}$) generated during the 3D reconstruction process.

ID	Diameter from Metallography (d) (mm)	Diameter from Calculation ($d_{com}$) (mm)	Error of Diameter (%)	Welding Depth from Metallography (h) (mm)	Welding Depth from Calculation ($h_{com}$) (mm)	Error of Welding Depth (%)
I	3.41	4.24	24.3	1.31	1.68	28.2
II	3.16	2.96	6.3	1.07	1.39	30.0
III	3.98	4.32	7.9	1.66	1.88	13.3
IV	3.48	3.60	3.4	1.25	1.59	27.2
V	3.63	3.60	0.8	1.38	1.68	21.7

## Data Availability

Data will be made available on request.

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
