# Peer review of "A Weld Pool Morphology Acquisition and Visualization System Based on an In Situ Calibrated Analytical Solution and Virtual Reality"

_sensors, 2025, doi:10.3390/s25092711_

Round 1

Reviewer 1 Report

Comments and Suggestions for Authors

First, I would like to briefly characterize the presented work. It is highly relevant and of great practical interest, as it is aimed at solving the problem of welding seam modeling in virtual reality. This task can be useful, among other things, for remote control or monitoring of welding processes, but not in a two-dimensional image format, but in a three-dimensional virtual space. This is undoubtedly the novelty of this work.
The work structure is standard and generally accepted. The methodology section is quite detailed and contains a large amount of graphic explanatory material, as well as an analytical model proposed by the authors for calculating weld parameters.
Perhaps the authors should expand the list of references, since their number is not very high and there are quite a lot of them with a date older than 5 years. 
The results are shown quite clearly. The main comments on this work are listed below:

1. It seems to me that a separate and more extensive Discussion section would strengthen the article. In it, I would like to see the reflections and recommendations of the authors on the delay between the actual state on the welding bath and the state displayed in the VR helmet; estimates of how large the error obtained in the study is or its level is acceptable for visualization and remote welding. 
2. Have there been any attempts to use not only an analytical model, but also any machine learning algorithm to predict weld parameters? Perhaps this would allow us to get interesting results. As a last resort, it's worth mentioning this as part of the Discussion.
3. The authors mention that the speed is limited by the hardware and the complexity of the model used. I would like more information about how this problem will be overcome, and the prospects for its solution (multithreading, using GPU for computing?). In addition, I did not see the characteristics of the current equipment, which did not provide the expected performance. 
4. As I understand it, the analytical model allows you to simulate a 3D visualization of a welding point with a high degree of reliability. Nevertheless, an error of 20-30% was obtained during the experiment. Does this error relate specifically to the calculations of the analytical model, or did other steps, including visualization, affect this error? What scenarios do the authors envisage for improving accuracy? (this is also good discussion material).

Thus, the work is certainly interesting, but it requires some refinement in terms of discussing the results obtained.

Author Response

Comment 1: It seems to me that a separate and more extensive Discussion section would strengthen the article. In it, I would like to see the reflections and recommendations of the authors on the delay between the actual state on the welding bath and the state displayed in the VR helmet; estimates of how large the error obtained in the study is or its level is acceptable for visualization and remote welding.

Response 1: Thank you for pointing this out. We agree with this comment. Therefore, we have added a new discussion section (Chapter 5 Discussion), which ranges from line 325 to 348. We mainly discussed the accuracy and time delay of the entire weld pool data acquisition system. We explored the main reasons for the errors in the 3D model and proposed corresponding solutions from line 326 to line 335, and we believe that it will be of greater help for welders to judge the teleoperated welding process if the error of the three-dimensional model can be controlled within 10%. And discussed the issue of time delay, namely the thoughts and suggestions you put forward regarding the delay between the actual state on the weld groove and the state displayed in the VR helmet from line 336 to line 348. The time delay can be reduced within 1 second by using the latest graphics card RTX 5090, which is acceptable.

Amendment 1: Lines 325-348:

  1. Discussion

Accuracy and time delay are two factors of the weld pool morphology acquisition and visualization system. In terms of accuracy, the precision of analytical calculation is a significant factor affecting the system's accuracy, while other processes, including the format conversion, packaging, and updating of the model, have no obvious impact on the model's accuracy. The average error of the three-dimensional model can be optimized by improving the analytical model itself, or by using more accurate data processing methods during the process of correcting the analytical model. For example, the deep learning network can be used to obtain a more accurate shape of the welding pool when processing the welding pool images. It will be of greater help for welders to judge the teleoperated welding process if the error of the three-dimensional model can be controlled within 10%.

In terms of time delay, there are two main factors in the weld pool data acquisition and visualization system that can lead to an increase in time delay. Firstly, the calculation time of the analytical model, which is approximately 5 seconds because the iterative time is set to 5 seconds (Δt in formula 3-2 and 3-3) considering the processing time of other processes. Secondly, the time spent on model packaging and transmission (2s), which is mainly related to the performance of the hardware equipment. The time delay can be reduced within 1 second by using the latest graphics card RTX 5090, whose computing performance is 20 times that of the equipment used in this paper (RTX 1060). The delay time of the entire system is taken as the longer value of the above two types of time delays by running the above two processes in parallel, which is about 5 seconds. The delay time can be optimized by improving the performance of the hardware equipment or decomposing the calculation process into multi-threaded synchronous calculations and using more computing devices for parallel computing.

Comment 2: Have there been any attempts to use not only an analytical model, but also any machine learning algorithm to predict weld parameters? Perhaps this would allow us to get interesting results. As a last resort, it's worth mentioning this as part of the Discussion.

Response 2: Thank you for pointing this out. The main purpose of using the analytical model in this paper is to reproduce the morphology of the underneath surface of the welding pool, which is different from the current conventional use of machine learning algorithms. Most of the current related studies focus on obtaining the weld morphology after welding is completed, or on constructing the morphology of the upper surface of the welding pool in real time during the welding process, and we have cited some recent literatures on the research of 3D morphology reconstruction and the application of neural networks (Ref. 14, 15, 22, 23, from line 41 to 49, and line 56 to 61) in this regard to better illustrate this point. These literatures also serve as a supplement to the issue you initially mentioned about the relatively few and outdated references in this paper.

In fact, machine learning algorithms can also be applied to the data acquisition and processing system constructed in this paper. For example, in the aspect of processing welding pool images, machine learning algorithms can also be used to optimize the processing procedure, so as to obtain more accurate parameters. This point has also been written into the main text of the article as part of the discussion (line 332 to 335), which is listed in the amendment of Response 1.

References:Ref. 14, 15, 22, 23

  • Li, B.; Xu, Z.; Gao, F.; Cao, Y.; Dong, Q. 3D Reconstruction of High Reflective Welding Surface Based on Binocular Structured Light Stereo Vision. Machines 2022, 10, 159. https://doi.org/10.3390/machines10020159
  • Gu, Z., Chen, J. & Wu, C. Three-Dimensional Reconstruction of Welding Pool Surface by Binocular Vision. J. Mech. Eng. 34, 47 (2021). https://doi.org/10.1186/s10033-021-00567-2
  • Li, L.D.; Cheng, F.J.; Wu, S.J. An LSTM-based measurement method of 3D weld pool surface in GTAW. Measurement 2021, 171, 108809. https://doi.org/10.1016/j.measurement.2020.108809
  • Gao, X.; Liang, Z.M.; Zhang, X.M.; Wang, L.W.; Yang, X. Penetration state recognition based on stereo vision in GMAW process by deep learning. Journal of Manufacturing Processes 2023, 89, 349-361. https://doi.org/10.1016/j.jmapro.2023.01.058

Amendment 2: Lines 41 to 49

Li used a binocular vision camera to construct a hole identification and two-way slicing methods based on 2D images, which can detect the weld morphology after welding (Ref. 14). Gu proposes the development of binocular vision algorithms to achieve real-time monitoring of the morphology of the upper surface of the welding pool (Ref. 15). The current teleoperation welding technology can only offer the welders merely two-dimensional (2D) images and average current/voltage, or the upper surface of the welding pool morphology. However, there is relatively little research on constructing the three-dimensional morphology of the underneath surface of the welding pool.

Lines 56 to 61:

Besides, Li proposes a novel method using the LSTM neural network to achieve the measurement of the 3D weld pool upper surface (Ref. 22). Xu explores a combination of two types of network strategies (computer stereo vision and classification decision), and adopts the supervised learning method to optimize the model for identifying the penetration state of the V-groove (Ref. 23).

Comment 3: The authors mention that the speed is limited by the hardware and the complexity of the model used. I would like more information about how this problem will be overcome, and the prospects for its solution (multithreading, using GPU for computing?). In addition, I did not see the characteristics of the current equipment, which did not provide the expected performance.

Response 3: Thank you for pointing this out. We agree with this comment. Therefore, we have focused on explaining how to overcome the issue of speed in the newly added discussion chapter (Chapter 5 Discussion), which is listed in Amendment 1. It is possible to increase the calculation speed by using equipment with higher performance or by using multiple devices for parallel computing (line 345 to 348).

Additionally, we have also specified the characteristics of the current equipment, including those of the computing device (line 249 to line 252) and the virtual reality device (line 227 to line 229).

Amendment 3: line 227 to line 229

The virtual environment device we used in the experiment is the HTC Vive Pro, which has an update frequency of 90Hz. Its display delay time is negligible compared with the delay times of analytical calculation and data transmission.

Line 249 to line 252

The analytical model of the welding temperature field is calculated in real time by a computer configured with an Intel Core i9 processor, 16 GB of running memory, and an Nvidia GeForce GTX 1060 graphics card.

Comment 4: As I understand it, the analytical model allows you to simulate a 3D visualization of a welding point with a high degree of reliability. Nevertheless, an error of 20-30% was obtained during the experiment. Does this error relate specifically to the calculations of the analytical model, or did other steps, including visualization, affect this error? What scenarios do the authors envisage for improving accuracy? (this is also good discussion material).

Response 4: Thank you for pointing this out. We agree with this comment. This error is indeed particularly related to the calculation of the analytical model, especially in the depth direction.

Besides, other steps, including visualization, have little impact on the model accuracy because these steps only involve format conversion and transmission of the model without any modifications.

In the discussion chapter, we believe that the error can be optimized by improving the analytical model itself, or by using more accurate data processing methods during the process of correcting the analytical model (line 330 to line 333), which is listed in Amendment 1. For example, the deep learning network can be used to obtain a more accurate shape of the welding pool when processing the welding pool images.

Amendment 4: line 330 to line 333

The content of Amendment 4 is included in Amendment 1.

Reviewer 2 Report

Comments and Suggestions for Authors

1.Enhance the visual elements of figures by adding markers and annotations to improve expressiveness. Ensure tables do not span pages, standardize borders, and adjust layouts to enhance formatting aesthetics.

2.Provide a comparative analysis of current 3D weld pool reconstruction methodologies, emphasizing distinctions between data-driven techniques and traditional physics-based models to highlight the technical advancement of the proposed analytical method.

3.Clarify the origin of material property values used in parameter tables by explicitly citing experimental datasets or prior publications, ensuring traceability for thermal/physical parameters.

4.Elaborate on computational hardware specifications and immersive visualization tools, including their performance limitations and influence on system latency during iterative processing stages.

5.Address systematic errors in weld pool depth predictions by analyzing potential causes such as geometric simplification assumptions, and propose targeted calibration strategies for future model improvements.

6.Standardize mathematical formulations across all temperature field equations, ensuring unified notation, proper dimensional consistency, and explicit declaration of physical variables.

7.Expand the literature survey to incorporate contemporary developments in 3D reconstruction technologies, particularly comparative evaluations of neural network approaches for dynamic welding process modeling.  

Author Response

Comment 1: Enhance the visual elements of figures by adding markers and annotations to improve expressiveness. Ensure tables do not span pages, standardize borders, and adjust layouts to enhance formatting aesthetics.

Response 1: Thank you for pointing this out. We agree with this comment. Therefore, we have made certain adjustments to the figures. We have adjusted the contrast between different texts in the figures by means of bolding. Additionally, we have enhanced the visual elements of some figures (Figure 2, 4, 6) through annotation. For example, we have annotated Figure 2(e) (The process of calibrating the heat source parameters) and Figure 2(f) (The process of image processing) to enable readers to have a better understanding in Figure 2. In terms of the tables, we have adjusted the context before and after to ensure that the tables do not span across pages. Thank you again for your suggestions.

Amendment 1: Figure 2, 4, 6(The original picture can be found in the attached Word document. )

Comment 2: Provide a comparative analysis of current 3D weld pool reconstruction methodologies, emphasizing distinctions between data-driven techniques and traditional physics-based models to highlight the technical advancement of the proposed analytical method.

Response 2: Thank you for pointing this out. We agree with this comment. Therefore, we have newly retrieved some research papers on the current 3D welding pool reconstruction methods (Ref. 14, 15, 22, 23) to highlight the technical advancement of the proposed analytical method. Traditional methods (including neural networks) have significant advantages in obtaining the surface morphology of the weld seam after welding or the upper surface morphology of the welding pool during the welding process. In contrast, the data - driven techniques in this paper has unique advantages in calculating the morphology of the underneath surface of the welding pool, which has hardly been mentioned in previous studies (line 41 to line 49 and line 56 to line 61).

Amendment 2: Lines 41 to 49

Li used a binocular vision camera to construct a hole identification and two-way slicing methods based on 2D images, which can detect the weld morphology after welding (Ref. 14). Gu proposes the development of binocular vision algorithms to achieve real-time monitoring of the morphology of the upper surface of the welding pool (Ref. 15). The current teleoperation welding technology can only offer the welders merely two-dimensional (2D) images and average current/voltage, or the upper surface of the welding pool morphology. However, there is relatively little research on constructing the three-dimensional morphology of the underneath surface of the welding pool.

Lines 56 to 61:

Besides, Li proposes a novel method using the LSTM neural network to achieve the measurement of the 3D weld pool upper surface (Ref. 22). Xu explores a combination of two types of network strategies (computer stereo vision and classification decision), and adopts the supervised learning method to optimize the model for identifying the penetration state of the V-groove (Ref. 23).

References:Ref. 14, 15, 22, 23

  • Li, B.; Xu, Z.; Gao, F.; Cao, Y.; Dong, Q. 3D Reconstruction of High Reflective Welding Surface Based on Binocular Structured Light Stereo Vision. Machines 2022, 10, 159. https://doi.org/10.3390/machines10020159
  • Gu, Z., Chen, J. & Wu, C. Three-Dimensional Reconstruction of Welding Pool Surface by Binocular Vision. J. Mech. Eng. 34, 47 (2021). https://doi.org/10.1186/s10033-021-00567-2
  • Li, L.D.; Cheng, F.J.; Wu, S.J. An LSTM-based measurement method of 3D weld pool surface in GTAW. Measurement 2021, 171, 108809. https://doi.org/10.1016/j.measurement.2020.108809
  • Gao, X.; Liang, Z.M.; Zhang, X.M.; Wang, L.W.; Yang, X. Penetration state recognition based on stereo vision in GMAW process by deep learning. Journal of Manufacturing Processes 2023, 89, 349-361. https://doi.org/10.1016/j.jmapro.2023.01.058

Comment 3: Clarify the origin of material property values used in parameter tables by explicitly citing experimental datasets or prior publications, ensuring traceability for thermal/physical parameters.

Response 3: Thank you for pointing this out. We agree with this comment. We have attached the relevant explanation above Table 2 (line 249). The materials in this article are the same as those in Reference 27, so the data in the parameter table are also consistent with those in Reference 27.

Amendment 3: Lines 249

The physical parameters of the material used in the analytical calculations are shown in Table 2, which is sourced from Ref. 27.

Comment 4: Elaborate on computational hardware specifications and immersive visualization tools, including their performance limitations and influence on system latency during iterative processing stages.

Response 4: Thank you for pointing this out. We agree with this comment. Therefore, we have added descriptions of relevant hardware specifications (lines 249 - 251) and visualization tools (lines 227 - 229) in the corresponding parts of the article.

In addition, we discussed the limitations of the equipment on the time delay in the newly added discussion chapter. the latency of the iterative processing on the system is described in lines 336 to 348.

Amendment 4: line 227 to line 229

The virtual environment device we used in the experiment is the HTC Vive Pro, which has an update frequency of 90Hz. Its display delay time is negligible compared with the delay times of analytical calculation and data transmission.

Line 249 to line 252:

The analytical model of the welding temperature field is calculated in real time by a computer configured with an Intel Core i9 processor, 16 GB of running memory, and an Nvidia GeForce GTX 1060 graphics card.

Line 336 to line 348:

In terms of time delay, there are two main factors in the weld pool data acquisition and visualization system that can lead to an increase in time delay. Firstly, the calculation time of the analytical model, which is approximately 5 seconds because the iterative time is set to 5 seconds (Δt in formula 3-2 and 3-3) considering the processing time of other processes. Secondly, the time spent on model packaging and transmission (2s), which is mainly related to the performance of the hardware equipment. The time delay can be reduced within 1 second by using the latest graphics card RTX 5090, whose computing performance is 20 times that of the equipment used in this paper (RTX 1060). The delay time of the entire system is taken as the longer value of the above two types of time delays by running the above two processes in parallel, which is about 5 seconds. The delay time can be optimized by improving the performance of the hardware equipment or decomposing the calculation process into multi-threaded synchronous calculations and using more computing devices for parallel computing.

Comment 5: Address systematic errors in weld pool depth predictions by analyzing potential causes such as geometric simplification assumptions, and propose targeted calibration strategies for future model improvements.

Response 5: Thank you for pointing this out. However, the key point of this paper is to use the analytical model and apply it in the context of remote welding. Therefore, not too much background information is elaborated. The derivation process of the analytical model used in this paper is described in detail in Reference 27, which also explains in detail the geometric simplification assumptions and other related contents of this analytical model.

Some research content on how to improve the model accuracy has been added (lines 327 - 335) in the newly added discussion section of this paper. Research methods such as neural networks can be used to improve the accuracy of the model. Thank you again for your suggestions to us.

Amendment 5: lines 327 to 335

In terms of accuracy, the precision of analytical calculation is a significant factor affecting the system's accuracy, while other processes, including the format conversion, packaging, and updating of the model, have no obvious impact on the model's accuracy. The average error of the three-dimensional model can be optimized by improving the analytical model itself, or by using more accurate data processing methods during the process of correcting the analytical model. For example, the deep learning network can be used to obtain a more accurate shape of the welding pool when processing the welding pool images. It will be of greater help for welders to judge the teleoperated welding process if the error of the three-dimensional model can be controlled within 10%.

Comment 6: Standardize mathematical formulations across all temperature field equations, ensuring unified notation, proper dimensional consistency, and explicit declaration of physical variables.

Response 6: Thank you for pointing this out. We agree with this comment. Therefore, we have made some modifications to Equation 2 and Equation 3 to ensure that their dimensions and physical variables are consistent.

Amendment 6:

Comment 7: Expand the literature survey to incorporate contemporary developments in 3D reconstruction technologies, particularly comparative evaluations of neural network approaches for dynamic welding process modeling.

Response 7: Thank you for pointing this out. We agree with this comment. Therefore, we have newly retrieved some research papers on the current 3D welding pool reconstruction methods, added some latest references (Ref. 14, 15, 22, 23), and described them in the chapter1 (line 41 to 49 and line 56 to 61). Citation 23 is a method of using the LSTM neural network to construct the upper surface of the welding pool during the dynamic welding process. The list of references is listed in Response 2.

Amendment 7: line 41 to 49 and line 56 to 61

Please refer to Amendment 2 for details.

Round 2

Reviewer 2 Report

Comments and Suggestions for Authors

All of my concerns have been addressed.